# Examining the Science Design Skills Competency among Science Preservice Teachers in the Post-COVID-19 Pandemic Period

Tafirenyika Mafugu 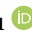

School of Education, Durban University of Technology, Durban 4001, South Africa; TafirenyikaM@dut.ac.za

**Abstract:** The study aimed to investigate the competencies of 42 preservice science teachers from a rural university in crafting scientific investigations while utilising the constructivist learning theory as its theoretical foundation. Employing an explanatory sequential design, the research initially collected quantitative data through a pre-test, followed by an intervention in the experimental group, succeeded by a post-test phase. In order to enrich the quantitative findings, qualitative data in the form of student responses were analysed, offering a more comprehensive understanding of the preservice teachers' proficiency in experimental design. Within the control group, no statistically significant variance emerged between the pre-test and post-test rankings, $Z = -1.3$, $p = 0.190$. In contrast, the experimental group exhibited noteworthy divergence. The Wilcoxon signed-rank test unveiled a substantial upsurge in post-test rankings when juxtaposed with the pre-test standings, $Z = -4$, $p < 0.001$. The qualitative data revealed that preservice teachers manifested a lack of familiarity with comprehension of the variables, strategies for ensuring investigation validity and reliability, and a coherent approach to gauging intervention impact. Emphasising the significance of these revelations, the study suggests plausible pathways for addressing these knowledge gaps via collaborative interventions, thus striving to effectively bridge the identified disparities.

**Keywords:** preservice teachers; explanatory sequential design; experimental group; control group; qualitative data

## 1. Introduction

The onset of the COVID-19 pandemic brought many challenges in the development of scientific competencies and skills. Most scientific skills need hands on experience in order for the skills to develop effectively. However, due to the onset of the COVID-19 pandemic, students at higher institutions were deprived of the need for hands-on exposure due to the need for social distancing that was imposed to limit the spread of COVID-19. A lack of hands-on experience could have an impact on the natural and applied sciences, which are rapidly growing occupational sectors in many countries, including the United Kingdom [1]. Hands-on practical activities are essential, as they assist in developing critical thinking skills that assist in effectively solving the multifaceted problems encountered in life [2]. Immediate solutions are often required for the problems we face, making it necessary to equip science students with scientific process skills that set them apart [3]. Hence, there is a need to compensate for losses in practical skills incurred during the COVID-19 pandemic. Science process skills encompass various abilities, such as equipment handling, observation, measurement, data collection, prediction, and hypothesis testing [4]. These skills play a vital role in developing problem-solving, critical thinking, and communication skills, which are highly valued in the future job market [5]. Possessing such skills empowers students to find solutions, interpret information, and effectively communicate their findings, all of which are key drivers of global economies [5,6]. Moreover, students proficient in science process skills contribute to generating new knowledge that is crucial for solving societal issues [5].

Engaging students in experiments to develop their scientific skills not only motivates them but also allows for first-hand experience and a deeper understanding of scientific phenomena [7]. Such experiences facilitate long-term retention of information, ensuring that learning becomes more permanent. The science curriculum has the responsibility of fostering science process skills to enhance scientific literacy among students [8,9]. These skills aid students in logical thinking and questioning.

Science process skills are fundamental in cultivating problem-solving abilities through the observation and practice of scientific phenomena [10,11]. Teachers play a crucial role in nurturing these skills among learners, making it essential for science preservice teachers to acquire the necessary scientific skills during their studies. Teachers have a significant impact on imparting scientific and problem-solving skills to students at an early stage, contributing to their social and cognitive development. Well-equipped teachers with developed science process skills are able to design appropriate investigations, even in under-resourced schools, to promote the development of these skills in students. It is essential to establish the foundation of problem-solving skills at an early age [11], as it positively impacts all areas of development, particularly the social and cognitive aspects, in children. When individuals acquire advanced problem-solving skills from a young age, they adapt more easily to society and their environment. The acquisition of such skills may have been hindered by the COVID-19 pandemic, which forced a shift to emergency remote teaching and disrupted laboratory practices worldwide. The shift to mandatory online teaching and learning posed a significant challenge for pre-service teachers, as it limited their ability to develop crucial skills through hands-on activities. It underscored the importance of digital literacy among pre-service teachers. There was a need for training and support in using various digital tools and platforms effectively. While hands-on activities were limited, pre-service teachers had to get creative with virtual alternatives. They had to share some innovative solutions and tools they used to provide interactive learning experiences, such as virtual labs, simulations, or collaborative online projects. However, the extent to which they developed resilience, flexibility, and innovative solutions to enable them to acquire crucial scientific skills is not clear. Online learning might have prevented preservice teachers from acquiring essential skills [12]. Limited information is available on students who were taught online regarding their proficiency in science process skills and whether they differ from students taught in traditional settings [13,14].

Stylinski et al. [15] underscores the fact that science skill assessment remains a significantly underutilised practice within the realm of science projects. In their literature review study, it became evident that merely a handful of respondents from the questionnaire and interview sessions, as well as a limited number of submitted articles, showcased any evidence of incorporating science skill assessment for learners. Furthermore, Burgess et al. [16] similarly identified sporadic instances of science skill assessment efforts. Moreover, the study conducted by Phillips et al. [17] indicated that the measurement of science inquiry skills ranked as the least commonly undertaken evaluation.

Multiple frameworks have been established to outline the essential competencies that higher education students should cultivate [18]. These competencies are presented in Table 1.

**Table 1.** Definition of competencies by national and international organisations (adapted from Vazquez-Villegas et al., [5]).

| Organisation | Competencies |
|---|---|
| National Research Council Framework (USA) [19] | Asking questions and defining problems<br>Developing and using models<br>Planning and carrying out investigations<br>Analysing and interpreting data<br>Using mathematics<br>Computational thinking<br>Constructing explanations<br>Designing solutions<br>Engaging in arguments from evidence<br>Obtaining, evaluating, and communicating Information |
| United Nations Educational, Scientific, and Cultural Organisation (UNESCO) [20] | Cognitive<br>Information processing (data interpretation and analysis)<br>Problem-solving<br>Engineering thinking<br>Scientific investigation<br>Computational thinking<br>Design thinking, creativity, and innovation |
| Next Generation Science Standards (USA) [21] | Asking questions and defining problems<br>Developing and using models<br>Planning and carrying out investigations<br>Analysing and interpreting data using mathematical and computational thinking<br>Constructing explanations and designing solutions<br>Engaging in arguments using evidence<br>Obtaining, evaluating, and communicating Information |

*1.1. The Impact of Activities Based on Science Process Skills on the Development of Problem-Solving Skills*

In a study conducted by Gültekin and Altun [6], it was found that activities focusing on science process skills significantly contributed to the development of problem-solving abilities. Their experimental group, which received various science process skills activities, demonstrated notable improvement compared to the control group, which did not receive such intervention. Furthermore, Artun et al. [22] emphasise the importance of science activities that are enriched with virtual reality technology, as it enables access to unobservable or dangerous phenomena. Virtual reality technology holds great potential in facilitating the acquisition of science process skills among secondary school learners and pre-service science teachers [23].

Several studies indicate that blended learning and experiment-based learning significantly improve critical and creative thinking [24]. Experiment-based learning offers numerous advantages, as highlighted by Ummah et al. [25]. These advantages include the enhancement of students' abilities. Experiment-based learning, for instance, provides students with opportunities to showcase their higher cognitive abilities. By engaging in experimental activities that require critical thinking and problem-solving, students can demonstrate their full potential through inquiry-based approaches, whereby students actively seek solutions by asking questions. Furthermore, Ummah et al. [25] highlight that experimental activities serve as a powerful motivator for students. By connecting classroom learning to real-life situations, it creates a sense of purpose and relevance, thereby increasing student engagement and enthusiasm. Additionally, Coyne et al. [26] assert that involving experimental activities in learning accommodates the needs of students with varying skills and learning styles. It provides a flexible and adaptable learning environment that can be customised to meet the unique requirements of each student, promoting

inclusivity and personalised learning experiences [26]. Moreover, it facilitates the design of plans and promotes the ability to articulate and explain the main questions of a project. This allows learners to shape their learning experiences [27]. Furthermore, experimental-based learning has additional strengths, which include enhancing critical thinking and problem-solving skills, developing personal communication skills, promoting information and media literacy, enabling teamwork and leadership abilities, encouraging creativity, and fostering innovation [27]. The outcomes of experimental-based learning implementation demonstrate that students who engage in such experiments exhibit improved collaboration, creative thinking, patience, and willingness to support and explain concepts to one another [28]. Anazifa and Djukri [29] suggest that experimental-based learning has a positive impact on students' creativity, as it involves the process of sensing, observing problems, generating hypotheses, testing them, and delivering results. It is emphasised that the understanding of the learning process and the interconnection between learning and creativity play a crucial role in enhancing creativity. Rahardjanto and Fauzi [30] assert that project activities encourage students to apply their creativity to solving real-world problems and leveraging their knowledge and skills to produce unique and innovative outcomes. Overall, studies support the idea that experimental-based learning offers a valuable framework for promoting active learning, critical thinking, collaboration, and creativity among students. By engaging in project-based activities, students have the opportunity to develop essential skills and to apply their knowledge in practical and meaningful ways. Experimental-based learning offers an alternative approach to cultivating essential scientific skills. Students could be tasked with at-home experiments, allowing for skill development even outside the traditional classroom. Nevertheless, the viability of this method for honing experimental design skills is somewhat uncertain. This is because it often necessitates specialised equipment and chemicals—resources that are not readily available in a typical home setting.

The study is grounded in the social constructivist theory, which perceives learning as an active process of constructing scientific knowledge through the social interaction between learners and their science teachers [31]. In this study, knowledge construction occurs through interactions among teachers, students, and the learning environment, including various learning platforms. As the learning process unfolds, assessment plays a pivotal role in evaluating the effectiveness of pedagogical strategies in science and gauging students' comprehension of the scientific content [32]. The feedback and interactions stemming from individual assessment activities facilitate the overall science learning process. Pedagogical instruction and assessment as underscored by Ghaica [33] are interdependent processes that are crucial for imparting scientific knowledge and providing support. Continuous assessment that encompasses both formative and summative approaches can furnish valuable feedback for identifying students who may be at risk by monitoring progress, adjusting pedagogical practices, and granting certification [33]. Moreover, assessment aids lecturers in reflecting upon their teaching methods, fostering student–lecturer discussions, and devising strategies to enhance teaching, learning, and academic achievement. As highlighted by Jimaa [34], it is worth noting that the choice of assessment methods significantly influences the learning process itself; learners tend to tailor their study approach based on the assessment requirements.

### 1.2. Study Context

The research was conducted at a rural university where the implementation of online learning was solely prompted by the emergence of the COVID-19 pandemic. Virtual laboratories had not been used to impart science process skills within the rural institution. Teachers were not familiar with the virtual laboratories, and the institution did not subscribe to any virtual laboratories because all experiments that were conducted before the pandemic were conducted through hands-on activities. The failure to make a transition to virtual laboratories could have significantly affected the acquisition of science process skills by the students, most of whom who came from underprivileged communities. The study

specifically centres around fourth-year science methodology students who were introduced to scientific concepts and science process skills during the pandemic. These students should eventually possess a comprehensive understanding of all science process skills, and the science methodology lecturers are responsible for imparting science pedagogical skills to them.

The onset of the COVID-19 pandemic presented unprecedented challenges to the development of science process skills. With face-to-face lessons suspended to minimise the spread of the virus, the traditional practical activities conducted in laboratories, which are essential for skill development, were no longer available. This absence of hands-on learning might have had a significant impact on the development of science process skills, which can lead to various challenges in the learning process. To address these challenges, this study aimed to investigate the development of science process design skills and proposed an intervention strategy. By focusing on students who were particularly affected by the shift to virtual learning, the study sought to provide practical recommendations to educators and practitioners on how to help these students catch up with their peers in terms of skill development. In conclusion, the disruption caused by the pandemic necessitated a deeper understanding of the impact on science education, especially regarding the acquisition of crucial science process skills. Through this research, we hoped to offer valuable insights and guidance to educators that might ensure that students can continue to develop these essential skills despite the challenges posed by virtual learning. The study sought to answer the following research question:

What are the preservice science teachers' experimental design competencies in the post-COVID-19 pandemic period?

## 2. Research Methodology

### 2.1. Objectives and Hypothesis of the Study

The study aimed to determine the impact of an intervention on preservice teachers' experimental design competencies in the post-COVID-19 pandemic period.

**Hypothesis 1.** *The mean ranks for the pre-test scores for the control and experimental groups are not statistically significantly different.*

**Hypothesis 2.** *The mean ranks for the post-test scores for the control and experimental groups are not statistically significantly different.*

### 2.2. Research Design and Participant Selection

The study followed an explanatory sequential design where quantitative data were collected from a sample of 42 preservice science teachers from a single institution situated in South Africa. There were 24 (57%) females, all of whom were in the age group of 22 to 27 years, and 18 (43%) males in the age group of 23 to 29 years. The majority (80%) of the participants were in the age group of 23 to 25 years. The participants were divided into two groups: an experimental group and a control group, each comprising 21 members. The participants were allocated to these groups through a random assignment process. Random numbers were generated within the required range and assigned to the respective control and experimental groups by utilising a random number generator (Figure 1).

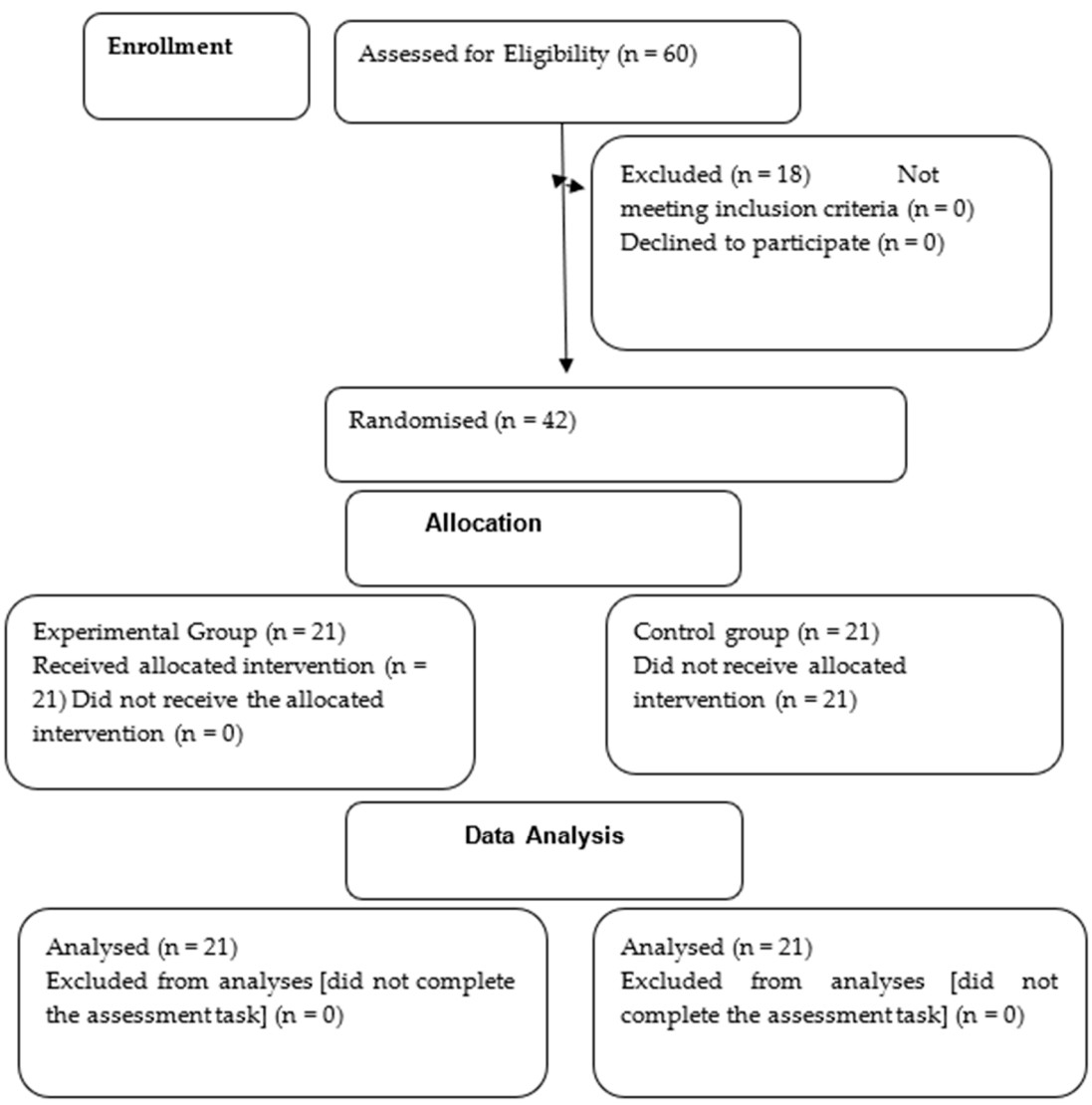

**Figure 1.** Flow diagram of the sampled participants.

*2.3. Data Collection Instruments*

The two groups of students were given a pre-test where they had to answer the following question individually in 30 min in the Blackboard learning management system:

*Birds and humans have similar pancreas tissues, with the same cell types contributing to exocrine and endocrine roles. Growth hormone (GH) is a peptide hormone that stimulates cell reproduction and regeneration in humans and other animals. It is produced during development to increase bone size and density.*

*GH can also be used in farming to enhance yields from different animals.*

*Design an experiment to investigate the following hypothesis:*

"Varying the concentration of GH injected affects the yield of meat from farmed chickens."

The responses were marked using the memorandum in Appendix A. An expert in the field was then given the responses to moderate the work. Following this assessment, an intervention was exclusively conducted for the experimental group. This intervention revolved around collaborative group assignments and was promptly succeeded by comprehensive classroom discussions.

The assignments encompassed three distinct tasks:

1. Formulate an experimental protocol to examine the impact of altitude variations on the quantity of red blood cells.
2. Devise an investigative approach to determine the concentration of an unidentified sample of reducing sugar.
3. Construct a comprehensive investigation to ascertain the correlation between temperature fluctuations and enzyme activity.

In the final investigation, students were equipped with essential materials such as water, potatoes, cork borers, test tubes accompanied by delivery tubes, hydrogen peroxide, and a water bath complete with a thermostat for precise temperature control. The materials were used to determine the effect of temperature on the activity of the enzyme peroxidase, which breaks down hydrogen peroxide into water and oxygen. In groups of five students each, the groups had to devise their own methodology. They demonstrated their methodology to the class, and the demonstrations were followed by class discussions. The discussions were conducted in weekly sessions of 1.5 h each over a period of three weeks. After three weeks, the post-test was administered. The post-test was similar to the pre-test question but was slightly modified. The post-test question is presented in Appendix B. The control group was then given the same treatment for the experimental group after the post-test to ensure that they were not disadvantaged. All the data collection process and investigations were carried out during extra time and not during normal class time.

*2.4. Quantitative Data Analysis*

The quantitative data were analysed using IBM SPSS Statistics version 29, with a significance level set at 5%. For the final pre-test marks, the Shapiro–Wilk test was performed to determine whether the data were normally distributed. This was carried out to determine the type of test to use in the comparison. Subsequent to the Shapiro–Wilk test, the results were analysed using the non-parametric equivalence of the independent sample t-test, the Mann–Whitney U test, to assess the baseline proficiency of both cohorts in the realm of experimental design processes. The non-parametric test was used since the data were not normally distributed, as was revealed by the Shapiro–Wilk test. The post-test results were compared with the pre-test results using the Wilcoxon test. The results are presented in tables and texts.

*2.5. Qualitative Data Analysis*

The qualitative data in the form of the answers presented by each student were collected and analysed thematically to diagnose the specific deficiencies of the answers provided by various students. The qualitative data generated from the students' answers were analysed manually using thematic analysis where themes were generated from the existing data (inductive). Coding and recoding were completed several times using the data from students' responses until the researcher came up with the final codes. Three themes emerged from the student's responses: lack of awareness regarding variables, lack of knowledge on how to ensure validity and reliability in the design, and failure to indicate the method and instrument to measure the impact. Trustworthiness in the qualitative data was ensured by analysing all responses provided until data saturation was reached, determined as being when no new themes emerged from the data set.

## 3. Results

Both quantitative and qualitative data are presented in this section.

*3.1. Pre-Test SPSS Output Comparing the Control and Experimental Groups*

The first section presents the pre-test data comparing the scores for the experimental and control groups. This section involved tested the following hypotheses:

**Hypothesis 1.** *The pre-test scores for the control group are normally distributed.*

**Hypothesis 2.** *The pre-test scores for the experimental group are normally distributed.*

**Hypothesis 3.** *The post-test scores for the control group are normally distributed.*

**Hypothesis 4.** *The post-test scores for the experimental group are normally distributed.*

The Shapiro–Wilk test was carried out to assist in determining whether the data were normally distributed. This stage is important, as it helps determine whether to use parametric or non-parametric tests. All Shapiro–Wilk test values were below 0.05, indicating that the data did not follow a normal distribution (Table 2). Therefore, the non-parametric Mann–Whitney and Wilcoxon tests were used in the comparisons.

**Table 2.** Shapiro–Wilk test for normality results.

|  | **Statistic** | **df** | **Sig.** |
|---|---|---|---|
| Pre-test control group | 0.844 | 20 | 0.004 |
| Pre-test experimental group | 0.767 | 20 | <0.001 |
| Post-test control group | 0.836 | 20 | 0.003 |
| Post-test experimental group | 0.878 | 20 | 0.017 |

*3.2. Pre-Test Results for the Control and Experimental Group*

The Mann–Whitney U Test was used to compare the mean (mean ranks) score for the control and the experimental group to determine whether the initial conditions for the two groups were the same. It was anticipated that the initial conditions would need to be almost the same to enable one to determine the impact of the intervention in the experimental group.

The pre-test scores for the experimental group were higher than the pre-test scores for the control group (Table 3).

**Table 3.** Mann–Whitney U-test mean and sum of ranks.

|  | **Groupings** | **n** | **Mean Rank** | **Sum of Ranks** |
|---|---|---|---|---|
| | Pre-test control group | 21 | 18.31 | 384.50 |
| Marks obtained | Pre-test experimental group | 21 | 24.69 | 518.50 |
| | Total | 42 | | |

From these data, it can be concluded that the mean scores for the control group and the experimental group were not statistically significantly different (U = 153, $p$ = 0.074) (Table 4).

**Table 4.** Mann–Whitney U-test statistics comparing pre-test results for the control and experimental groups.

|  | **Marks Obtained in the Pre-Test** |
|---|---|
| Mann–Whitney U | 153.500 |
| Wilcoxon W | 384.500 |
| Z | −1.787 |
| Asymp. Sig. (2-tailed) | 0.074 |

*3.3. The Pre- and Post-Test SPSS Outputs Comparing the Pre-Test and Post-Test Scores for Both the Experimental Group and the Control Group*

In this section, the pre- and post-test results for both the control and the experimental group were compared. This was done to determine whether the intervention had any

impact on the experimental group. Table 5 shows the mean and standard deviation of the scores.

**Table 5.** Descriptive statistics for the pre-tests and post-tests for the control and experimental groups.

|  | n | Mean | Std. Deviation | Minimum | Maximum |
|---|---|---|---|---|---|
| Pre-test control group | 21 | 1.81 | 0.928 | 0.00 | 3.00 |
| Pre-test experimental group | 21 | 2.29 | 0.784 | 0.00 | 3.00 |
| Post-test control group | 21 | 2.10 | 0.995 | 1.00 | 5.00 |
| Post-test experimental group | 21 | 6.52 | 1.470 | 2.00 | 9.00 |

Std. = standard.

In the control group, three post-test scores showed a decrease compared to the corresponding pre-test scores, whereas six post-test scores exhibited an increase (Table 6). The remaining 12 post-test scores in the control group remained unchanged from the pre-test scores. Conversely, in the experimental group, only 1 post-test score was lower than the pre-test score, with a notable 20 post-test scores displaying an improvement over the pre-test scores.

**Table 6.** Mean and sum of ranks for Wilcoxon signed-rank test.

|  |  | n | Mean Rank | Sum of Ranks |
|---|---|---|---|---|
| Post-test—pre-test control groups | Negative ranks | 3 [a] | 4.00 | 12.00 |
|  | Positive ranks | 6 [b] | 5.50 | 33.00 |
|  | Ties | 12 [c] |  |  |
|  | Total | 21 |  |  |
| Post-test—pre-test experimental groups | Negative ranks | 1 [d] | 1.00 | 1.00 |
|  | Positive ranks | 20 [e] | 11.50 | 230.00 |
|  | Ties | 0 [f] |  |  |
|  | Total | 21 |  |  |

[a] Post-test control < pre-test control. [b] Post-test control > pre-test control. [c] Post-test control = pre-test control. [d] Post-test experimental < pre-test experimental. [e] Post-test experimental > pre-test experimental. [f] Post-test experimental = pre-test experimental.

In the control group, there was no statistically significant difference observed between the pre-test and post-test ranks, as evidenced by the Z score of $-1.3$ and *p*-value of 0.190 (Table 7). Conversely, within the experimental group, the Wilcoxon signed-rank test revealed a significant increase in post-test ranks compared to pre-test ranks, with a Z score of $-4.01$ and a *p*-value of < 0.001.

**Table 7.** Test statistics for the Wilcoxon signed-rank tests.

|  | Post-Test Control Group—Pre-Test Control Group | Post-Test Experimental Group—Pre-Test Experimental Group |
|---|---|---|
| Z | $-1.310$ [b] | $-4.012$ [b] |
| Asymp. Sig. (2-tailed) | 0.190 | <0.001 |

[b] Based on negative ranks.

## 4. Qualitative Data (Pre-Intervention)

This section presents the qualitative data to enrich the quantitative data that were collected. The data were collected through the analysis of the answers provided by students to identify the specific issues with which the learners needed more assistance. The data from all students were analysed manually using thematic analysis, with themes generated from the existing data (inductive) and data saturation reached when there were no new themes

emerging from the data. Representative answers for students were selected purposefully by choosing representative answers that best represented the theme under consideration.

**Theme 1: Lack of Awareness Regarding Variables**

In various scenarios, the absence of awareness concerning pertinent variables was observed in various responses provided by the participants. The lack of understanding of the independent and controlled variables was reflected in the responses provided by some participants. One participant presented the following responses to the question:

Participant ST1 incorrectly stated time as the independent variable and GH concentration as the controlled variable, clearly indicating a lack of awareness regarding variables (Figure 2). The lack of knowledge of the variables was also evident in the response provided by ST2 (Figure 3) in the following response to the question:

Apparatures
1. growth hormones
2. chickens
3. time
4. syringe
variables
independent variable: time
dependent variable: growth of chickens
controlled variable: GH concentration
experiment steps
question: what are the effects of HG on the yield meat?
hypothesis
the meat yield differs depending on the amount of GH injected
experiment
use syringe to inject 5g of GH on chicken
use syringe to inject 10g of GH on other chicken
do the same thing on other chicken using 15g of GH
check the results after 3 weeks
results
the chicken with higher GH concentration grew more than others
conclusion
the amount of GH influences the growth of the yield chicken

**Figure 2.** Reflecting the lack of awareness regarding variables (ST1).

Experiment
Independent variable = Meat
Dependent variable= concentration
Controlled variable = chickens
Hypothesis= Differing the concentration of Growth Hormone injected affects the yield of meat from farmed chickens.

**Figure 3.** Reflecting the lack of awareness regarding variables (ST2).

In addition to the lack of independent and controlled variables reflected in STI, ST2 also failed to identify the dependent variable (Figure 3). The student identified "concentration" as the dependent variable instead of "yield of chicken."

## 5. Lack of Validity and Reliability in the Design

The lack of knowledge on how to ensure validity and reliability in the investigation was reflected in the above responses from ST1 and ST2. The participants were unable to list all controlled variables that were to be considered in the investigation. Furthermore, the issue of ensuring reliability was not mentioned at all. The student failed to indicate the need to use more than one chicken for each concentration of growth hormone and to determine the change by finding the average weight for the number of chickens in each group. This lack of knowledge was further seen in the response given by ST3 (Figure 4):

Aim: To investigate whether the growth hormone injected in chickens affect the yield of meat from farmed chicken.
Hypothesis: Varying the concentration of growth hormone injected affects the yield of meat from farmed chickens.
Materials:
1. Growth hormone injection.
2. Two chickens.
3. Scale

Methods:
Inject one of the two chickens with the grown hormone injection and put them in the same place to observe the results.
Results:
After 5.5 weeks we measured the two chickens and found that the chicken that was injected weights 2,900Grams and the

chicken that was not injected weights 940 Grams.

**Figure 4.** Lack of knowledge of validity in the design (ST3).

## 6. Failure to Indicate the Method and Instrument to Measure the Impact

The responses given by ST1, ST2, and ST3 also reflected the ignorance of the criteria to measure the impact on the concentration of the growth hormone. Although STI indicated that chickens with higher concentrations of growth hormone would grow faster, they did not indicate how the growth was to be measured in the investigation. Additionally, ST4 indicated that the measurement would be completed but did not mention the need to measure the growth at certain time intervals (Figure 5). Participant ST4 did not justify the need to slaughter the chickens and measure the weight eventually, or that the measurement would initially be taken in live chickens.

- Pick 10 healthy chickens at random, all of the same breed, age, and weight range.
- Grouping: With a control group in mind, divide the chickens into five groups of 2 chickens each.
- GH Injection: While the other four groups will each receive an injection of GH at a different concentration, the control group will just receive a saline solution injection. Throughout the lifespan of the chickens, the GH injection will be given at regular intervals in accordance with the dosage advised.
- Over the following six months, keep an eye on the chickens' development and weight gain.
- Meat yield measurement and slaughter: After six months, slaughter the birds and weigh the meat you've collected.
- Data Analysis: Analyze the data to determine if there is a correlation between the concentration of GH injected and the yield of meat from farmed chickens.
- Results: The results of the experiment will indicate whether or not varying the concentration of GH injected affects the yield of meat from farmed chickens.

**Figure 5.** Failure to indicate the method and instrument to measure the impact (ST4).

### 7. Qualitative Data (Post-Intervention)

A great improvement was observed in the responses given by the students in the experimental group three weeks after the intervention. The control group's response remained substandard after three weeks. Most of the issues that were lacking in the responses provided before the intervention were eliminated by the gain of knowledge during the intervention process. This improvement was evident in the responses given by ST2 after the intervention (Figure 6):

**Experimental Design:**
1. **Sample Selection:**
   - Randomly select a group of healthy and similar-sized rabbits from the farm population to ensure that the starting conditions are consistent across all subjects.
2. **Control Group:**
   - Have a control group of rabbits that receive a placebo or a normal saline solution without GH. This will serve as a baseline to compare the results obtained from the GH-treated groups.
3. **Experimental Groups:**
   - Divide the remaining rabbits into several experimental groups, each receiving a different concentration of GH (1%, 2%, 3%, 4%, and 5%).
4. **Administration of GH:**
   - Administer the appropriate GH concentration to each experimental group. This can be done via injections, and the injections should be given consistently and at the same time intervals throughout the experiment.
5. **Feeding and Care:**
   - Ensure that all rabbits, including the control group, receive the same standard diet and care to minimize any potential confounding factors related to nutrition and general health.
6. **Data Collection:**
   - Monitor the rabbits regularly and record relevant data, including weight gain, size, and overall health. Keep track of the amount of meat produced by each rabbit at the end of the experiment.
7. **Statistical Analysis:**
   - Analyze the data using appropriate statistical methods, such as ANOVA (Analysis of variance), to compare the meat yield between different GH concentrations and the control group.

**Figure 6.** Response of ST2 after the intervention.

A similar trend of improved understanding of concepts was also observed in the responses of ST4 after intervention (Figure 7). The responses from participant preservice teacher ST4 indicate a clear understanding of variables, issues of validity and reliability, and the need to measure weight. However, it was not necessary to kill the animal before measurement since all other measurements were taken using the live animal.

Independent: concentration of GH injected

dependent variables: rate of growth

Title: The effect of varying growth hormone concentrations on rabbit meat yield.

Objective: The main objective of this experiment is to determine if different concentration of growth hormone injections have an impact on the yield of meat from farmed rabbits.

Experimental Design:

1. Sample Selection

-Select a homogenous group of healthy rabbits from the same breed and age group.

2. Randomization

-Randomly divide the selected rabbits into five groups. Each group will receive a different concentration of GH injected as follows: 1%,2%,3%,4%,5% GH. Ensure that the groups have a similar distribution of age and weight.

3.GH administration

-Prepare GH solutions of 1%,2%,3%,4%, and 5%concentration.

-Administer the GH injections to each group according to their assigned concentration. The injection frequency and dosage should be based on existing scientific literature or expect recommendations to ensure safe and ethical practices.

4.Feeding and care

-House all rabbits in identical and appropriate living conditions, providing them with a standardized diet and water access.

-Regularly monitor the rabbit's health and well-being throughout the experiment.

5.Data collection

-After a predetermined period, measure the final weight of each rabbit in all groups

Euthanize the rabbits humanely, and then carefully separate and collect the meat from each rabbit

-Weigh the meat from each rabbit to determine the yield of meat for each group.

6.Data analysis

-Analyse the data using appropriate statistical methods, such as analysis of variance or t-tests to compare the meat yields between different GH concentration groups.

-Assess if there are any statistically differences in meat yield among the groups.

**Figure 7.** Responses of ST4 after the intervention.

## 8. Discussion

In various scenarios, the absence of awareness concerning pertinent variables can lead to significant challenges. When students lack understanding about the variables at play, it becomes difficult to make informed decisions or devise effective design strategies. Unawareness of variables can have adverse consequences across diverse domains. For instance, overlooking key variables in scientific research can undermine the validity of experimental results and hinder the progress of knowledge. To address this issue, it is crucial to emphasise a thorough analysis of the different variables in different experimental designs. By diligently identifying and assessing relevant variables in the different experimental designs and engaging students in group designs and class discussions, one can achieve a clearer understanding of the different variables to consider in the experimental design. In summary, recognising the pivotal role that variables play in any experimental design is essential. By actively seeking out, understanding, and incorporating these variables in design processes, students can enhance their overall effectiveness in the experimental design process. Valls-Bautista et al. [35] and García-Carmona et al. [36] also found that students had shortcomings in identifying variables and in planning the complete development of a guided scientific inquiry.

The study reflected that students lacked knowledge of how to ensure validity and reliability when planning practical investigations. They lacked understanding of the need to keep other variables constant to ensure that the dependent variable is only affected

by one variable at a time. This had the impact of hindering their ability to plan and design investigations in under-resourced rural environments, where most of them are often deployed after completing their studies. The study findings indicate inadequate knowledge of variables and issues of validity and reliability and concur with the findings of Ozer and Sarıbaş [37], which indicated that preservice teachers had an inadequate understanding of inquiry in some aspects, even after the treatment.

However, engaging students in collaborative hands-on activities and group work was observed to greatly improve the post-test scores of the experimental group, whereas the pre-and post-test scores of the control group were not significantly different. This significant improvement was also observed in the post-test answers presented by students. In Burke et al.'s [38] study, preservice teachers placed more importance on "collaboration with fellow teachers" and "student collaborative in science learning." The importance of collaboration seems to be confirmed in this study. Additionally, this study's findings are in line with the findings in Yustina et al.'s [24], and Valls-Bautista et al.'s [35] studies, which indicate that intervention strategies improve students' inquiry-based skills. However, Ozer and Sarıbaş [37] indicate that intervention strategies only improve some of the students' inquiry-based skills, especially if the intervention is administered over a short period of time. For greater skill development, several interventions need to be implemented. The existing literature highlights that the understanding of scientific inquiry (SI) and science practices (SPs) does not seamlessly transition into practical teaching methods [39,40]. This underscores the need for a comprehensive understanding of these elements throughout teacher training. Achieving this understanding could be facilitated through explicit inquiry-based instructional activities and active involvement in pedagogical inquiry pursuits [27,41], which were hindered during the COVID-19 pandemic. In the study by Burke et al. [38], both pre- and post-service teachers ranked investigation-based science through hands-on activities as being among the most critical topics for professional development in preservice teachers, as they could assist them to develop skills they could use in their classrooms.

Although this study had the strength of using both quantitative and qualitative data, the results were limited to only one institution with a small sample size of 42 students. It only focused on assessing the design skills. Future studies need to focus on multiple science skills in international studies with large sample sizes to yield more generalisable results.

### 9. Implications of the Study

The study underscores the presence of knowledge gaps in science process skills that have arisen due to reduced engagement in hands-on inquiry-based activities—a consequence of the COVID-19 pandemic. This research also emphasises the urgency of developing diverse interventions tailored to address this issue across all impacted student cohorts, spanning both tertiary and high school levels. Importantly, the study proposes potential avenues for mitigating these knowledge gaps through collaborative interventions, thereby aiming to bridge the identified disparities.

**Funding:** The research received no external funding.

**Institutional Review Board Statement:** The study was conducted in accordance with the Declaration of Helsinki, and approved by the Institutional Review Board of the University of the Free State (UFS-HSD2021/0351/22).

**Informed Consent Statement:** Informed consent was obtained from all subjects involved in the study.

**Data Availability Statement:** The data presented in this study are available on request from the corresponding author.

**Conflicts of Interest:** The author declares no conflicts of interest.

### Appendix A. Memorandum

Independent variable: concentration of growth hormone/reject GH/reject hormone (1 mark for the correct independent variable)

Dependent variable: mass of chicken/increase in weight/growth of chicken/reject chicken /weight (1 mark for the correct dependent variable)

Controlled variables: the amount of food/type of food/day-old chick/same breed/same space/same amount of water/same feeding time/same person to feed/same time of measurement/measurement taken by the same person using the same scale; accept any other reasonable answers (1 mark for any two correct answers, maximum points 3 marks)

A reasonable number of chickens injected for each concentration (1 mark) and average weight for the group (1 mark) over a long period/a month or more (1 mark)

Good sentence construction and coherence (2 marks)

An instrument for measurement mentioned (1 mark)

Frequency of measurement (1 mark)

Method of hormone administration mentioned (1 mark)

Any other reasonable answers

## Appendix B. Post-Intervention Question

Mammals and humans have similar pancreas tissues, with the same cell types contributing to exocrine and endocrine roles. Growth hormone (GH) is a peptide hormone that stimulates cell reproduction and regeneration in humans and other animals. It is produced during development to increase bone size and density.

GH can also be used in farming to enhance yields from different animals.

You are provided with 1%, 2%, 3%, 4%, and 5% GH. Design an experiment to investigate the following hypothesis:

> "Varying the concentration of GH injected affects the yield of meat from farmed rabbits."

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
