# Peer review of "Examining the Science Design Skills Competency among Science Preservice Teachers in the Post-COVID-19 Pandemic Period"

_education, doi:10.3390/educsci14040387_

Round 1

Reviewer 1 Report (Previous Reviewer 3)

Comments and Suggestions for Authors

The article reports on research the competences of trainee science teachers at a rural university in the development of scientific investigations in the post-COVID-19 period.

Although the introduction provides information on the impact of COVID-19 and remote teaching on preservice teachers' science competencies, the relationship with the results of the study is not apparent. How can it be ensured that it was remote teaching that impacted on these competencies? (14/468: “The study underscores the presence of knowledge gaps in science process skills, which have arisen due to reduced engagement in hands-on inquiry-based activities; a consequence of the COVID-19 pandemic”). The authors would have to explain this further, perhaps by providing information about the context of the rural university where the study took place.

Both research design and analysis are adequate, and the use of non-parametric tests is appropriate given the small sample size. It is suggested to further detail the population n of the sample, indicating gender, age, and any variables of interest to the study.  In the methodology section, it is also necessary to detail how the qualitative analysis was carried out: whether grounded theory was considered, whether the analysis was inductive or deductive, whether a computer programme was used for the analysis, etc.

 It is recommended to change the format of figures 2-7. Instead of providing the image with the response, it is preferable that the authors comment on the analysis performed.

Comments on the Quality of English Language

No comments.

Author Response

Reviewer 2 Report (Previous Reviewer 2)

Comments and Suggestions for Authors

Despite the relevance and interest of the proposal's theme, it was expected that the results would indicate a lack of pre-service teachers' skills. Therefore, the impact is not new. However, the proposal is balanced between a good theoretical foundation and an exhaustive data analysis. In technical terms, I have nothing to add, because its internal structure allowed us to draw conclusions based on the research carried out. I insist that what the authors found in the investigation carried out is not new.

Author Response

Reviewer 3 Report (New Reviewer)

Comments and Suggestions for Authors

Thank you for the opportunity to review this manuscript.  It has a potential to contribute to knowledge on preservice science teachers’ design process skills.

The authors need to provide clarity in terms of how intervention was conducted, details should include the duration of the intervention, was it a workshop, did it happen during the lecture times? What activities were given to the control group so that they benefit from this study?  

We are told on page 6 that “’The participants were divided into two groups: an experimental group and a control group, each comprised of 21 members.’’ What was the motivation to use the experimental and control group and how were the ethical issues addressed?  

The memorandum suggest that marks were allocated for each question. The test scores for individual participant for both pre and post test are not included in the report.  

There is also a need to provide some clarity on how the assignment was marked.

Round 2

Reviewer 1 Report (Previous Reviewer 3)

Comments and Suggestions for Authors

The authors have taken into account the reviewer's suggestions and have improved the wording of the article.

Comments on the Quality of English Language

No comments on the quality of English Language.

Reviewer 3 Report (New Reviewer)

Comments and Suggestions for Authors

The information of the actual percentage scores can be included as an appendix of the paper.   

This manuscript is a resubmission of an earlier submission. The following is a list of the peer review reports and author responses from that submission.

Round 1

Reviewer 1 Report

Comments and Suggestions for Authors

The article overall is good and provides information that is significant. 

What were some of the questions posed to participants in the qualitative piece?

It the article seemed disjointed and the results were not clear to me. I understand the knowledge gap in the science process skills and interventions. I think more work needs to be added haveing a conclusion and results clearly stated

Author Response

What were some of the questions posed to participants in the qualitative piece?

The qualitative section involved the analysis of the written responses by students and not interview questions. There were no interviews.

The article seemed disjointed, and the results were not clear to me. I understand the knowledge gap in the science process skills and interventions. I think more work needs to be added to have a conclusion and results clearly stated.

The reviewer seem not to be familiar with some statistical terms that were relevant in the analysis. Reviewer 2 confirmed this information.

Reviewer 2’s comments support that the paper is comprehensive.

Reviewer 2 Report

Comments and Suggestions for Authors

Despite the relevance and interest of the proposal's theme, it was expected that the results would indicate a lack of pre-service teachers' skills. Therefore, the impact is not new. However, the proposal is balanced between a good theoretical foundation and an exhaustive data analysis. In technical terms, I have nothing to add, because its internal structure allowed us to draw conclusions based on the research carried out. I insist that what the authors found in the investigation carried out is not new.

Author Response

Despite the relevance and interest of the proposal's theme, it was expected that the results would indicate a lack of pre-service teachers' skills. Therefore, the impact is not new. However, the proposal balances a good theoretical foundation and an exhaustive data analysis. In technical terms, I have nothing to add because its internal structure allowed us to draw conclusions based on the research carried out. I insist that what the authors found in the investigation carried out is not new

Reviewer 1 indicated the existence of the knowledge gap. Although the reviewer indicated that it is not new, this article reveals the extent of the gap that needs to be addressed in the context of the study.

Reviewer 3 Report

Comments and Suggestions for Authors

The study investigates the science design skill competences among science preservice teachers. Below are some comments to improve the readability of the article.

The abstract focuses too much on the results of the study, without placing the potential reader in the importance of the study, its justification, the sample (how many teachers have participated and where did they come from?), among others. The abstract should be modified to provide more data on the study. It is recommended to follow the journal template: The abstract should be a total of about 200 words maximum. The abstract should be a single paragraph and should follow the style of structured abstracts, but without headings: 1) Background: Place the question addressed in a broad context and highlight the purpose of the study; 2) Methods: Describe briefly the main methods or treatments applied. Include any relevant preregistration numbers, and species and strains of any animals used; 3) Results: Summarize the article's main findings; and 4) Conclusion: Indicate the main conclusions or interpretations. The abstract should be an objective representation of the article: it must not contain results which are not presented and substantiated in the main text and should not exaggerate the main conclusions.

With regard to the introduction, the authors provide context for the need for teachers to possess scientific competences and skills that they can promote among students. However, it is suggested that they revise the wording of the introduction, as it is unclear how it relates to the mandatory online teaching during the COVID-19 pandemic. The PjBL is also alluded to but fails to link it to the purposes of the study.

It is recommended that the Methodology section follows a more scientific structure: objectives of the study, participants (describing the sample, its selection - and providing as much detail as possible), how the 42 participants were chosen, randomly or by convenience; procedure, instruments used and everything related to the quantitative and qualitative analyses (programmes used, statistics, etc.).

It is recommended that the results section be redrafted and structured in a clearer way. It is very confusing and the reader does not understand what has been done and for what purpose. The results have to be related to the objectives of the study, which are absent.

The authors are recommended to revise the Discussion section, adding limitations to the study and future perspectives. The sample is only 42 teachers, too small to extend generalisations.

Comments on the Quality of English Language

No comments on the quality of English Language

Author Response

The abstract focuses too much on the results of the study, without placing the potential reader in the importance of the study, its justification, the sample (how many teachers have participated and where did they come from?), among others.

The issues in the abstract were checked and issues that were missing were added.

The abstract should be a single paragraph and should follow the style of structured abstracts, but without headings: 1) Background: Place the question addressed in a broad context and highlight the purpose of the study; 2) Methods: Describe briefly the main methods or treatments applied. Include any relevant preregistration numbers and species and strains of any animals used; 3) Results: Summarize the article's main findings; and 4) Conclusion: Indicate the main conclusions or interpretations. The abstract should be an objective representation of the article: it must not contain results which are not presented and substantiated in the main text and should not exaggerate the main conclusions.

The study aimed to investigate the competencies of 42 preservice science teachers from a rural university in crafting scientific investigations while utilising the constructivist learning theory.

The incomplete sentence “This was substantiated by a Z score of -1.3 and a.” was deleted from the abstract.

The length was adhered to.

Round 2

Reviewer 3 Report

Comments and Suggestions for Authors

With regard to the introduction, the authors provide context for the need for teachers to possess scientific competences and skills that they can promote among students. However, it is suggested that they revise the wording of the introduction, as it is unclear how it relates to the mandatory online teaching during the COVID-19 pandemic. The PjBL is also alluded to but fails to link it to the purposes of the study.

It is recommended that the Methodology section follows a more scientific structure: objectives of the study, participants (describing the sample, its selection - and providing as much detail as possible), how the 42 participants were chosen, randomly or by convenience; procedure, instruments used and everything related to the quantitative and qualitative analyses (programmes used, statistics, etc.).

It is recommended that the results section be redrafted and structured in a clearer way. It is very confusing and the reader does not understand what has been done and for what purpose. The results have to be related to the objectives of the study, which are absent.

The authors are recommended to revise the Discussion section, adding limitations to the study and future perspectives. The sample is only 42 teachers, too small to extend generalisations.

Comments on the Quality of English Language

No comments.

Author Response

I have tried to make additions to the introductions, restructure the methods section, add limitations and future forecast in an effort to overcome the weaknesses highlighted by the reviewer. Some statements were added in the results section to assist the reader to understand better.